# Calibration and analysis of genome-based models for microbial ecology

**Stilianos Louca[1]\*, Michael Doebeli[2]**

[1]Institute of Applied Mathematics, University of British Columbia, Vancouver, Canada; [2]Department of Zoology, University of British Columbia, Vancouver, Canada

**Abstract** Microbial ecosystem modeling is complicated by the large number of unknown parameters and the lack of appropriate calibration tools. Here we present a novel computational framework for modeling microbial ecosystems, which combines genome-based model construction with statistical analysis and calibration to experimental data. Using this framework, we examined the dynamics of a community of *Escherichia coli* strains that emerged in laboratory evolution experiments, during which an ancestral strain diversified into two coexisting ecotypes. We constructed a microbial community model comprising the ancestral and the evolved strains, which we calibrated using separate monoculture experiments. Simulations reproduced the successional dynamics in the evolution experiments, and pathway activation patterns observed in microarray transcript profiles. Our approach yielded detailed insights into the metabolic processes that drove bacterial diversification, involving acetate cross-feeding and competition for organic carbon and oxygen. Our framework provides a missing link towards a data-driven mechanistic microbial ecology.

## Introduction

Metabolic interactions are an emergent property of microbial communities (*Morris et al., 2013*; *Chiu et al., 2014*). Even the simplest life forms can only be understood in terms of biological consortia characterized by shared metabolic pathways and distributed biosynthetic capacities (*Klitgord and Segrè, 2010*; *McCutcheon and Moran, 2012*; *Husnik et al., 2013*). For example, glucose catabolism to carbon dioxide or methane is a multi-step process often involving several organisms that indirectly exchange intermediate products through their environment (*Stams, 1994*). Microbial communities are thus complex systems comprising several interacting components that cannot be fully understood in isolation. In fact, metabolic interdependencies between organisms are at least partially responsible for our current inability to culture the great majority of prokaryotes (*Schink and Stams, 2006*). Understanding the emergent dynamics of microbial communities is crucial to harnessing these multicomponent assemblages and using synthetic ecology for medical, environmental and industrial purposes (*Brenner et al., 2008*).

Genome sequencing has enabled the reconstruction of full-scale cell-metabolic networks (*Henry et al., 2010*), which have provided a firm basis for understanding individual cell metabolism (*Varma and Palsson, 1994*; *Duarte et al., 2004*; *Klitgord and Segrè, 2010*). Recent work indicates that multiple cell models can be combined to understand microbial community metabolism and population dynamics (*Stolyar et al., 2007*; *Klitgord and Segrè, 2010*; *Zengler and Palsson, 2012*; *Chiu et al., 2014*; *Harcombe et al., 2014*). These approaches assume knowledge of all model parameters such as stoichiometric coefficients, maintenance energy requirements or extracellular transport kinetics, a requirement that is rarely met in practice (*Feist et al., 2008*; *Harcombe et al., 2014*). Experiments and monitoring of environmental samples could provide valuable data to calibrate microbial community models, for example, via statistical parameter estimation, but appropriate tools are lacking. So far, the standard approach has been to obtain each parameter through laborious specific measurements or from the available literature, or to manually adjust

**\*For correspondence:** stilianos.
louca@gmail.com

**Competing interests:**
See page 14

**Reviewing editor**: Wenying Shou, Fred Hutchinson Cancer Research Center, United States

**eLife digest** Microbes like bacteria and yeast play important roles in the environment, human health and even some industrial processes. However, it is difficult to understand the roles of microbes in these situations because many different types of microbes often live together in complex communities. Some of the microbes may compete with each other for resources like oxygen or sugar. Others may rely on one another for survival. For example, one microbe may feed on molecules that are released as waste from another microbe.

To better understand these microbial communities, we first need to understand the processes by which each microbe uses nutrients and releases waste molecules that influence other microbes. Researchers have used a technique called 'genome sequencing' to reconstruct the networks of genes and chemical reactions that are involved in these processes, and to build computer models of microbial communities in different environments.

However, the existing models can be labor intensive and do not allow researchers to easily use statistics to analyse them. To address this problem, Louca and Doebeli created a new computer model with built-in statistical tools that accurately predicts the interactions in communities that contain multiple strains of a bacterium called *Escherichia coli.* First, Louca and Doebeli grew a single strain of *E. coli* in the laboratory for many generations, which led to the evolution of the bacteria so that two new strains emerged. One of the new strains was more efficient at using sugar as a food source than the other and sometimes released a molecule called acetate. The other new strain became more efficient at using this acetate.

Next, Louca and Doebeli used data that had been collected for each individual strain, to test whether the model could recreate the way that the new strains had evolved together. The model accurately predicted that the two new strains would gradually replace the original strain. The strain that was more efficient at using sugar emerged first, which led to extra acetate being available for the other new strain that became more efficient at using acetate.

Louca and Doebeli's findings demonstrate for the first time that data collected for individual microbes can be used to explain the dynamics and evolution of small communities of microbes using mathematical models. The next step is to test this approach on larger communities in the environment.

parameters to match observations (*Mahadevan et al., 2002*; *Chiu et al., 2014*; *Harcombe et al., 2014*). Furthermore, statistical model evaluation and sensitivity analysis is typically performed using ad hoc code, thus increasing the effort required for the construction of any new model. Consequently, the experimental validation of genome-based microbial community models and their application to biological questions are rare (*Meadows et al., 2010*; *Harcombe et al., 2014*).

We have developed MCM (Microbial Community Modeler), a mathematical framework and computational tool that unifies model construction with statistical evaluation, sensitivity analysis and parameter calibration. MCM is designed for modeling multi-species microbial communities, in which the metabolism and growth of individual cell species is predicted using genome-based metabolic models. Cells in the community interact in a dynamical environment in which metabolite concentrations and other environmental variables influence, and are influenced by, microbial metabolism. Unknown model parameters can be automatically calibrated (fitted) using experimental data such as cell densities, nutrient concentrations or rate measurements. To demonstrate the potential of MCM, we modeled a bacterial community that has emerged from in vitro evolution experiments, during which an ancestral strain repeatedly diversified into two distinct ecotypes. Experiments with microbes have an established tradition as model systems for understanding ecological and evolutionary processes (*Elena and Lenski, 2003*; *Kassen and Rainey, 2004*). We show that the predictions derived from MCM are in very good agreement with the outcomes of several monoculture and coculture experiments. While the experimental results described below have been found over the course of several years (*Friesen et al., 2004*; *Spencer et al., 2007*; *Le Gac et al., 2008*; *Herron and Doebeli, 2013*), it is only now that a mechanistic model has managed to unify them in a clear, unambiguous and synergistic manner. The analysis presented here thus provides strong credence to a large body of experimental work that was done in our lab over the course of roughly a decade.

## Model

In MCM, a microbial community model is a set of differential equations for the population densities of the cell species comprising the community and of the ambient concentrations of utilized nutrients (metabolites), coupled to optimization problems for the cell-specific rates of reactions involving these metabolites. Each cell is characterized by its metabolic potential, that is, the genetically determined subset of reactions it can catalyze, as well as any available metabolite transport mechanisms. The reaction rates and metabolite exchange rates (i.e. the metabolism) of each cell are assumed to depend on its metabolic potential as well as on the current environmental conditions, such as metabolite concentrations. Through their metabolism, in turn, cells act as sinks and sources of metabolites in the environment. Additional metabolite fluxes, such as oxygen diffusion from the atmosphere into the growth medium of a modeled bacterial culture, can be included in the model.

At any point in time, individual cell metabolism is determined using flux balance analysis (FBA) (*Orth et al., 2010*), a widely used framework in cell-metabolic modeling (*Varma and Palsson, 1994*; *Duarte et al., 2004*; *Klitgord and Segrè, 2010*; *Freilich et al., 2011*; *Chiu et al., 2014*). In FBA, cell metabolism is assumed to be regulated in such a way that the rate of biosynthesis is maximized (*Varma and Palsson, 1994*; *Feist and Palsson, 2010*). The chemical state of cells is assumed to be steady, leading to stoichiometric constraints that need to be satisfied for any particular combination of intracellular reaction rates. Reaction rates, on the other hand, are limited due to finite enzyme capacities. Metabolite uptake/export rates can also be limited due to finite diffusion rates or limited transmembrane transporter efficiency. For example, uptake rates can be Monod-like functions of substrate concentrations (*Mahadevan et al., 2002*; *Harcombe et al., 2014*). Taken together, cell-metabolic potential, stoichiometric consistency, reaction rate limits and transport rate limits define the constraints of a linear optimization problem for each cell species at each point in time. The optimized biosynthesis rate is translated into a cell production rate by dividing by the cell's mass, thus defining the species' population growth (*Figure 1*).

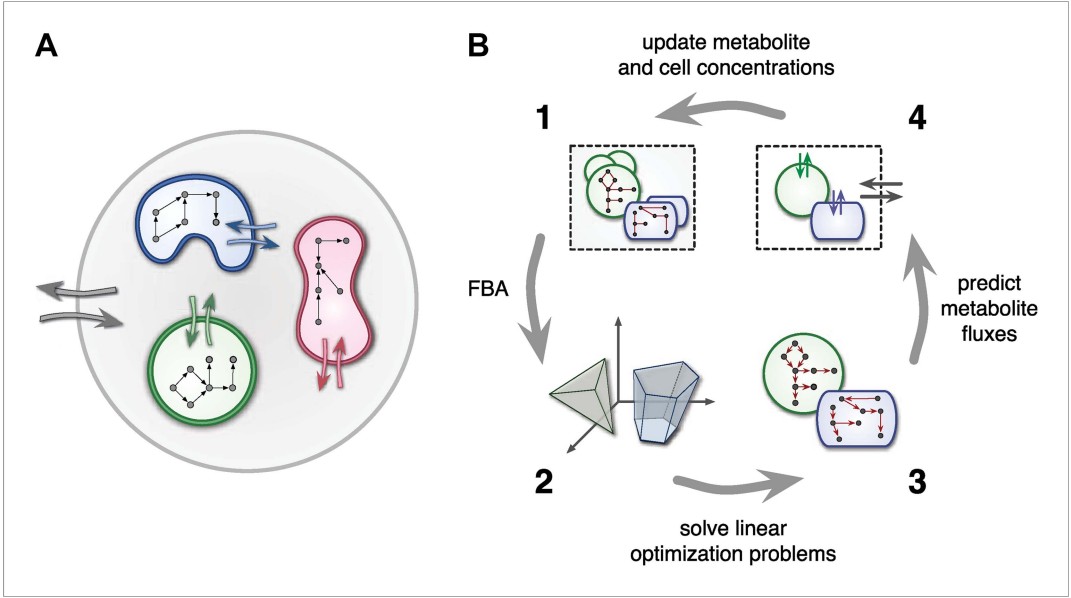

**Figure 1**. Framework used by MCM. (**A**) Conceptual framework used by MCM. Cells (colored shapes) optimize their metabolism for maximal growth and influence their environment via metabolite exchange (small colored arrows). Additional external fluxes can also affect the environment (large grey arrows). The environment, in turn, influences each cell's metabolism. (**B**) Computational framework used by MCM. Each iteration consists of four steps: flux balance analysis (FBA) is used to translate cell-metabolic potentials and environmental conditions (1) into a linear optimization problem for the growth rate of each cell species (2). The set of possible reaction rates corresponds to a polytope in high-dimensional space. Solving the optimization problems (3) yields predictions on microbial metabolite exchange rates (4). Metabolic fluxes and cell growth rates are used to predict metabolite and cell concentrations in the next iteration (1).

The central assumption of individual cells maximizing biosynthesis, subject to environmental and physiological constraints, is rooted in the idea that evolution has shaped regulatory mechanisms of unicellular organisms in such a way that they strive for maximum growth whenever possible. Biosynthesis has been experimentally verified as an objective for *Saccharomyces cerevisiae* and *E. coli* (*Burgard and Maranas, 2003*; *Gianchandani et al., 2008*; *Harcombe et al., 2013*). The assumption of maximized biosynthesis is less valid for genetically engineered organisms or those exposed to environments that are radically different from the environments that shaped their evolution (*Segrè et al., 2002*). Despite its limitations, FBA has greatly contributed to the understanding of several genome-scale metabolic networks and metabolic interactions between cells (*Stolyar et al., 2007*; *Klitgord and Segrè, 2010*; *Orth et al., 2010*; *Freilich et al., 2011*; *Chiu et al., 2014*; *Harcombe et al., 2014*). One advantage of FBA models over full biochemical cell models is their independence of intracellular kinetics and gene regulation, which limits the number of required parameters to stoichiometric coefficients and uptake kinetics.

The combination of FBA with a varying environmental metabolite pool, as implemented by MCM, is known as dynamic flux balance analysis (DFBA) (*Mahadevan et al., 2002*; *Chiu et al., 2014*; *Harcombe et al., 2014*). In contrast to conventional FBA, DFBA models are dynamical because cell densities and environmental metabolite concentrations both change with time, and the rate of change of each cell density and metabolite concentration depends on the current cell densities and metabolite concentrations (*Mahadevan et al., 2002*; *Harcombe et al., 2014*). Because metabolites can be depleted or produced by several cell species, the environmental metabolite pool mediates the metabolic interactions between cells (*Schink and Stams, 2006*). For example, oxygen uptake rates might depend on environmental oxygen concentrations, which in turn are reduced by cellular respiration. Similarly, cells might excrete acetate as a byproduct of glucose catabolism, which then becomes available to other cells. The metabolic optimization of individual cells striving for maximal growth, while modifying their environment, leads to non-trivial community dynamics that can include competition, cooperation and exploitation. The cell-centric nature of DFBA differs fundamentally from other flux balance analyses of microbial communities that assume an optimization of a community-wide objective such as total biomass synthesis (*Stolyar et al., 2007*; *Klitgord and Segrè, 2011*; *Zomorrodi and Maranas, 2012*). Such an assumption is at least questionable from an evolutionary perspective and likely not appropriate for communities comprising several species (*Mitri and Foster, 2013*).

Recent work suggests that DFBA is a promising approach to microbial ecological modeling (*Meadows et al., 2010*; *Chiu et al., 2014*; *Harcombe et al., 2014*). For example, *Harcombe et al. (2014)* designed a computational tool (COMETS) based on DFBA, which was able to accurately predict equilibrium compositions of mixed bacterial cultures grown on petri dishes. However, COMETS offers limited model versatility in terms of uptake and reaction kinetics and only has few environmental feedback mechanisms (namely, varying extracellular metabolite concentrations). Furthermore, it assumes complete knowledge of all required model parameters and provides no generic statistical model analysis. Hence, while COMETS sets an important precedent, considerable work is still needed to make DFBA a practical approach in microbial ecosystem modeling. MCM extends Harcombe et al.'s framework to more versatile microbial ecological models that include arbitrary reaction kinetics (e.g., subject to product-inhibition) as well as dynamical environmental variables (e.g., pH) that influence, and are influenced by, microbial metabolism. In addition, MCM supports cell models in which internal molecules act as dynamical constraints that further restrict the FBA solution space, for example to account for post-transcriptional regulation or delays in enzyme synthesis (*Blazier and Papin, 2012*). These so called regulatory FBA models have been shown to improve the fidelity of conventional FBA models for *E. coli* and *S. cerevisiae* (*Covert et al., 2001*; *Covert and Palsson, 2002*; *Covert et al., 2004*; *Herrgård et al., 2006*), however their application to microbial communities remains untested. MCM can statistically evaluate models against data, analyze their sensitivity to varying parameters (*Cariboni et al., 2007*), and estimate the uncertainty of model predictions in the face of stochasticity (*Hammersley and Handscomb, 1964*). Perhaps most importantly, MCM can automatically calibrate unknown model parameters to data, for example obtained from monoculture experiments (as demonstrated below), from bioreactor experiments involving multiple species (*Louca and Doebeli, 2015*) or from environmental samples of unculturable communities (*Figure 2*; see the 'Materials and methods' and the Supplement for details). MCM can thus be used to understand the dynamics of realistic microbial ecosystems, ranging from the soil or groundwater to mixed laboratory cultures and bioreactors.

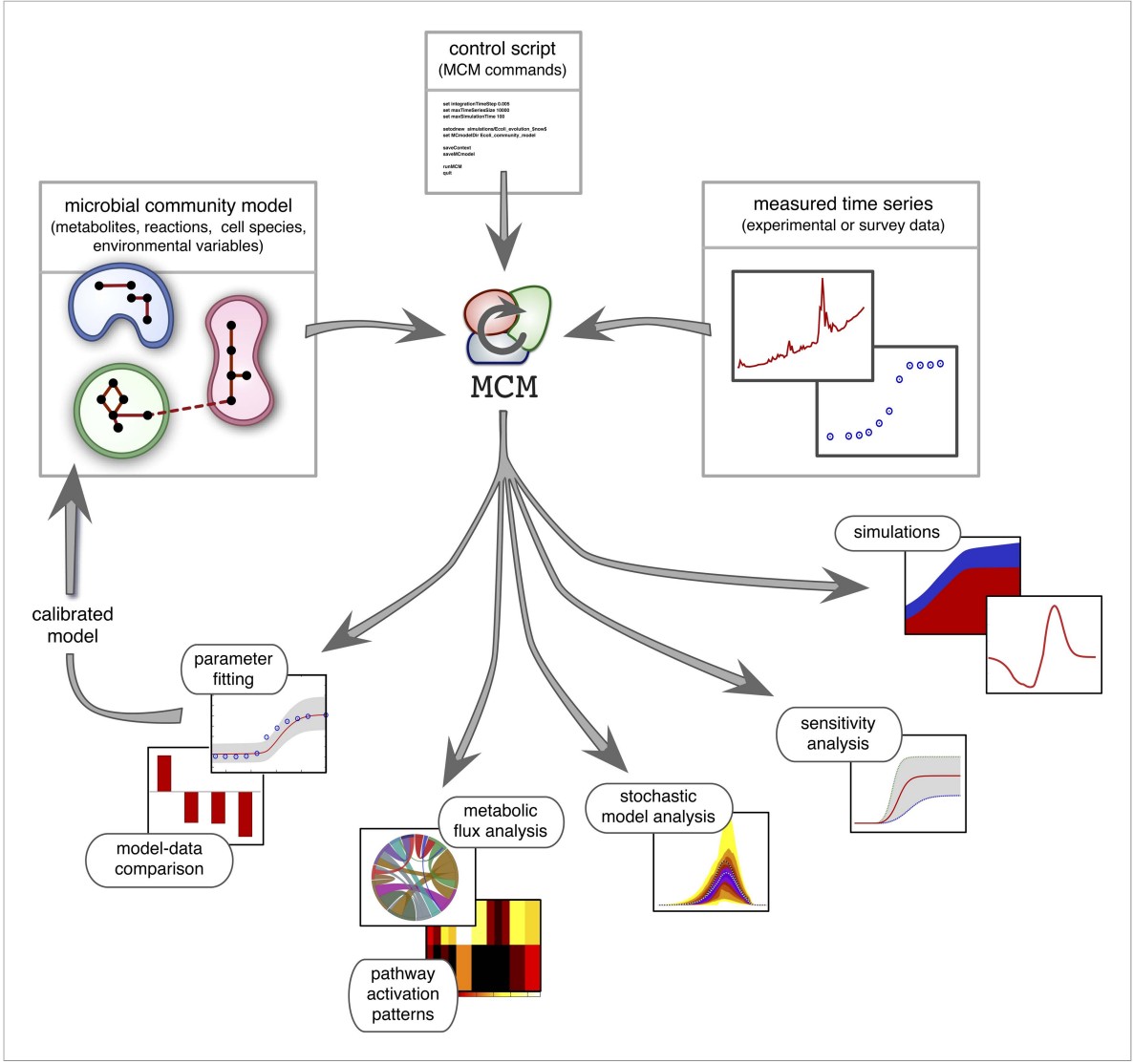

**Figure 2**. Overview of MCM's working principle and functionalities: A microbial community model is specified using human-readable configuration files in terms of metabolites, reactions, the metabolic potential of cell species and any additional environmental variables. Models with multiple ecosystem compartments are also possible. A script with MCM commands controls the analysis of the model and, if needed, its calibration using experimental data. The calibrated model can also be used to create new, more complex models (as exemplified in this article).

## Results and discussion

### Successional dynamics of a microbial community

In a series of laboratory evolution experiments with *E. coli* (strain B REL606; *Yoon et al., 2012*) in glucose-acetate supplemented medium, two metabolically distinct strains consistently evolved from the ancestral (A) strain (*Le Gac et al., 2008*; *Spencer et al., 2008*; *Herron and Doebeli, 2013*). When grown in monoculture with the same medium composition, all three strains exhibit diauxic growth curves with a fast glucose-driven growth phase followed by slower growth on acetate. However, the three strains differ in their efficiencies to catabolize glucose and acetate: Strain SS (slow switcher) is a better glucose utilizer when compared to strain A, and the depletion of glucose only leads to a slow switch to acetate consumption. On the other hand, the FS (fast switcher) strain has evolved to be a better acetate utilizer, initiating acetate consumption at higher remnant glucose concentrations than

strains A and SS. This acetate specialization is based on a tradeoff in the citric acid cycle and comes at the cost of being a less competitive glucose consumer.

Replicated serial dilution experiments starting with strain A monocultures have shown a consistent phenotypic diversification, involving an initial invasion of the SS phenotype and a subsequent invasion of the FS phenotype, leading to the eventual extinction or near-extinction of the ancestor and the stable coexistence of the SS and FS phenotypes (*Figure 3*) (*Le Gac et al., 2008*; *Spencer et al., 2008*; *Tyerman et al., 2008*; *Herron and Doebeli, 2013*). Genome sequencing revealed that this metabolic diversification can be attributed to point-mutations in genes linked to glucose and acetate uptake kinetics and metabolism (*Herron and Doebeli, 2013*). The successional dynamics of the three phenotypes are thus likely driven by adaptations to a changing metabolic niche space, defined by fluctuating glucose, acetate and, potentially, oxygen availabilities (*Le Gac et al., 2008*; *Tyerman et al., 2008*; *Herron and Doebeli, 2013*). An understanding of the underlying ecological processes would shed light on the ecology and evolution of natural microbial communities with shared catabolic pathways.

To mechanistically explain the observed community dynamics, we used MCM to construct a model comprising the ancestral and the two evolved *E. coli* types. By keeping track of pathway activation, cell densities, metabolic fluxes and nutrient concentrations, we gained detailed insight into the processes driving the successional dynamics of metabolic diversification.

## Experimental calibration

Based on a published cell-metabolic template for the ancestral *E. coli* strain comprising over 2000 reactions (*Yoon et al., 2012*), we first constructed three separate cell models for the phenotypes A, SS and FS, respectively. In these preliminary models, cells grew on a substrate pool that resembled previous batch-fed monoculture experiments with glucose-acetate supplemented minimal medium (*Le Gac et al., 2008*). Cell-specific oxygen, acetate and glucose uptake rate limits were Monod-like functions of substrate concentrations (*Emerson and Hedges, 2008*; *Millero, 2013*). We calibrated several physiological parameters for each cell type to measured chemical concentration and cell density profiles, using least squares fitting (*Figure 4*). MCM automatically calibrates free parameters to data through an optimization algorithm that involves step-wise exploration of parameter space and repeated simulations (see 'Materials and methods' and Supplementary Material).

We then constructed the microbial community (MC) model by combining the three calibrated cell models into a community growing in a common substrate pool. The environmental context resembles Herron & Doebeli's evolution experiments (*Herron and Doebeli, 2013*). In particular, the model includes realistic oxygen depletion-repletion dynamics (*Gupta and Rao, 2003*), glucose and acetate depletion by microbial consumption, as well as daily dilutions into fresh glucose-acetate supplemented medium at a factor 1:100. The microbial community initially consists mostly of type A ($10^{10}$ cells/l), while both SS as well as FS cells are assumed to be rare (1 cell/l). Because the model is

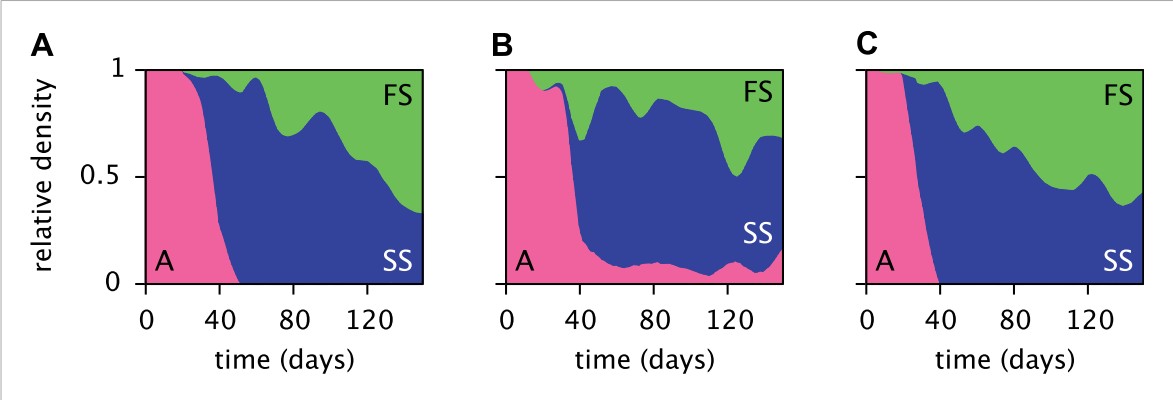

**Figure 3**. Estimated relative cell densities of the A, SS and FS types during three replicated evolution experiments by Herron and Doebeli (2013, *Figure 2*), starting with the same ancestral *E. coli* strain. Within each of the three experiments (**A**–**C**), the illustrated SS or FS lineage comprises several strains with varyingly pronounced SS or FS phenotypes, respectively. Cell generations were translated to days by assuming an average of 6.7 generations per day (*Herron and Doebeli, 2013*).

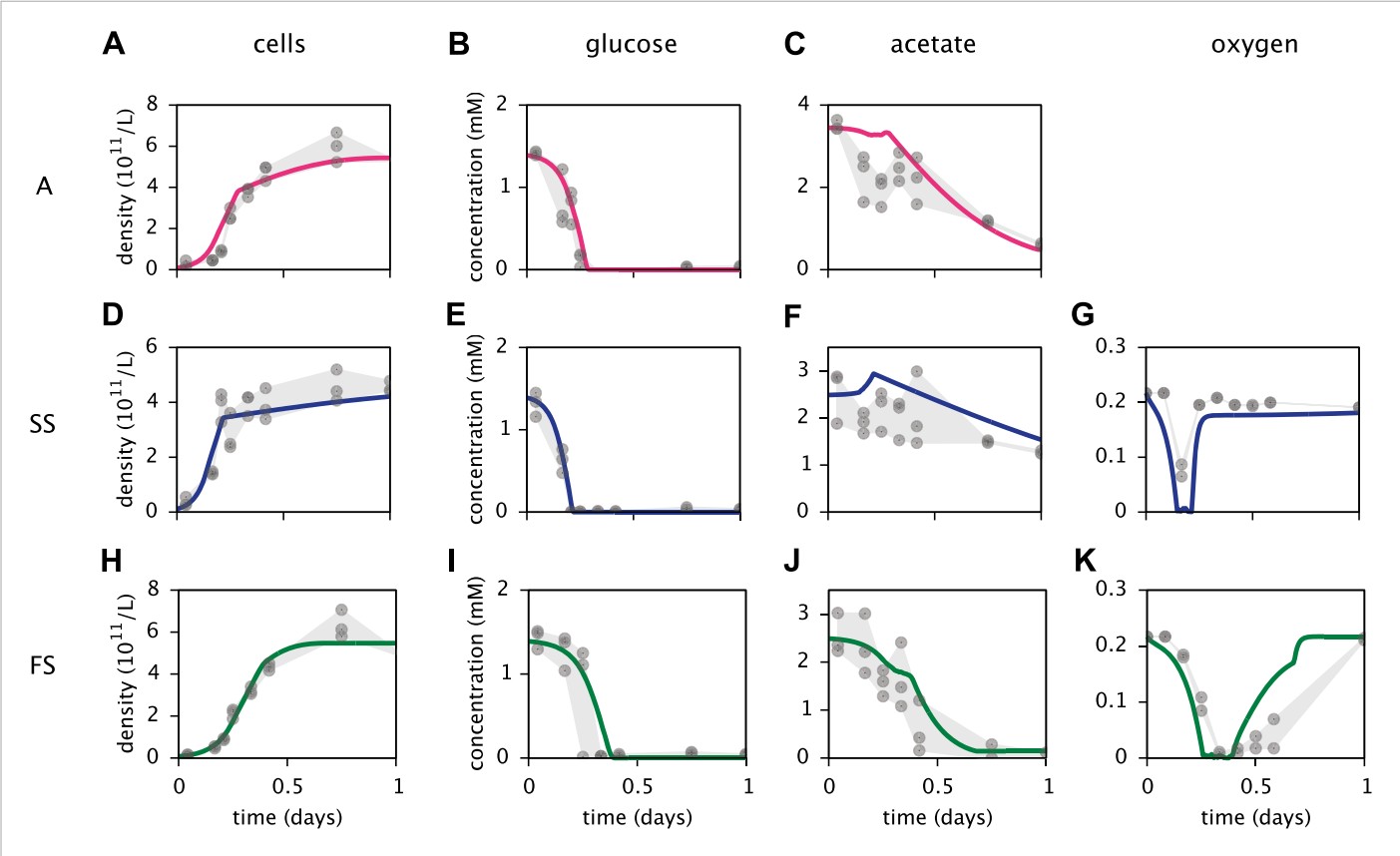

**Figure 4**. Calibration of *E. coli* cell models. Continuous curves: Time course of cell densities, glucose concentration, acetate concentration and oxygen concentration (columns 1–4, respectively) predicted by MCM for monocultures of strain A, SS and FS (rows 1–3, respectively) grown on glucose-acetate medium. Points are data used for model calibration, and were obtained from analogous monoculture growth experiments (*Le Gac et al., 2008*). Oxygen data were not available for strain A.

deterministic, the invasion or extinction of each type only depends on its growth rate in a possibly changing environment, but not on random mutation events, nor on demographic stochastic fluctuations.

## Predicting microbial community dynamics

Simulations of the MC model reproduced the successional dynamics observed in Herron & Doebeli's experiments: An initial replacement of the ancestor by the SS type is followed by an invasion of the FS type, leading to the eventual coexistence of the SS and FS types and extinction of the ancestral strain (*Figure 5A*). Interestingly, FS can also invade in the absence of SS, however invasion occurs much slower and FS reaches lower densities than in the presence of SS (*Figure 5—figure supplement 1*). This is consistent with an early presence of the FS lineage at low densities in the evolution experiments (*Figure 3*), indicating that some of the first FS mutations already confer a slight advantage over the ancestor when FS is rare (*Herron and Doebeli, 2013*).

Time series of acetate concentrations (*Figure 5B*) link the observed successional dynamics of the three types to a gradually changing metabolic niche space: The replacement of type A by the more efficient glucose specialist SS leads to an accumulation of acetate and facilitates the invasion of the FS type. The specialization of the SS and FS types on glucose and acetate, respectively (*Figure 6A*), enables their long-term coexistence on glucose-acetate enriched medium through frequency dependent competition (*Friesen et al., 2004*; *Le Gac et al., 2008*; *Herron and Doebeli, 2013*). In fact, cell-specific acetate exchange rates reveal that the SS type temporarily excretes acetate during short intervals, which is concurrently and subsequently consumed by the FS type (*Figure 5G*). This

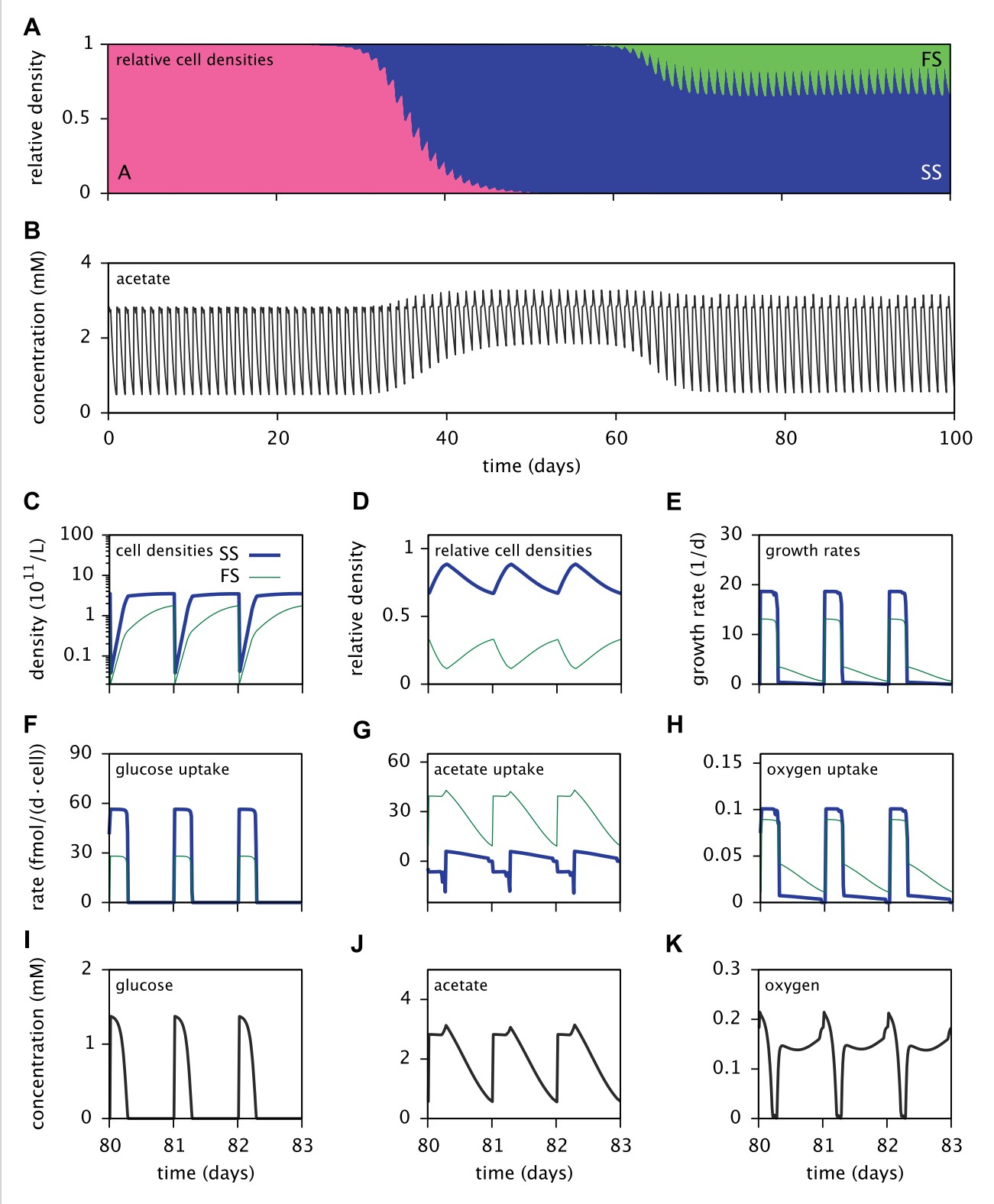

**Figure 5**. Dynamics of the *E. coli* microbial community model. (**A**) Relative cell densities of the A, SS and FS types over time. (**B**) Acetate concentration over time. (**C**), (**D**) and (**E**): SS and FS cell densities, relative cell densities and growth rates over time, respectively, during stable coexistence. (**F**), (**G**) and (**H**): Cell-specific glucose, acetate and oxygen uptake rates over time, respectively. Negative values correspond to export. (**I**), (**J**) and (**K**): Glucose, acetate
*Figure 5. continued on next page*

*Figure 5. Continued*

and oxygen concentrations over time, respectively. Diurnal fluctuations in all figures are due to daily dilutions into fresh medium. Tics on the time axes in (**C–K**) mark points of dilution.

The following figure supplements are available for figure 5:

**Figure supplement 1**. Predicted relative cell densities of the A and FS types in coculture, in the absence of SS.

**Figure supplement 2**. Robustness of the predicted stable coexistence of the SS and FS types in coculture.

periodic acetate cross-feeding is an evolutionarily emergent property of the microbial community (*Treves et al., 1998*). The temporary production of acetate by the SS type is consistent with previous SS-FS coculture experiments, which revealed slightly increased acetate concentrations towards the end of the SS exponential growth phase (*Spencer et al., 2007*). An evolved increase of acetate excretion by *E. coli* in glucose minimal medium has also been reported by *Harcombe et al. (2013)*.

It should be noted that cell metabolism depends on substrate concentrations and is subject to strong temporal variation. In particular, acetate excretion by SS cells correlates strongly with oxygen limitation (*Figure 5G,K*). The excretion of acetate by *E. coli* as a byproduct of oxygen-limited glucose catabolism has been observed experimentally and explained using flux balance analysis (*Mahadevan et al., 2002*). In the absence of oxygen limitation, complete aerobic glucose

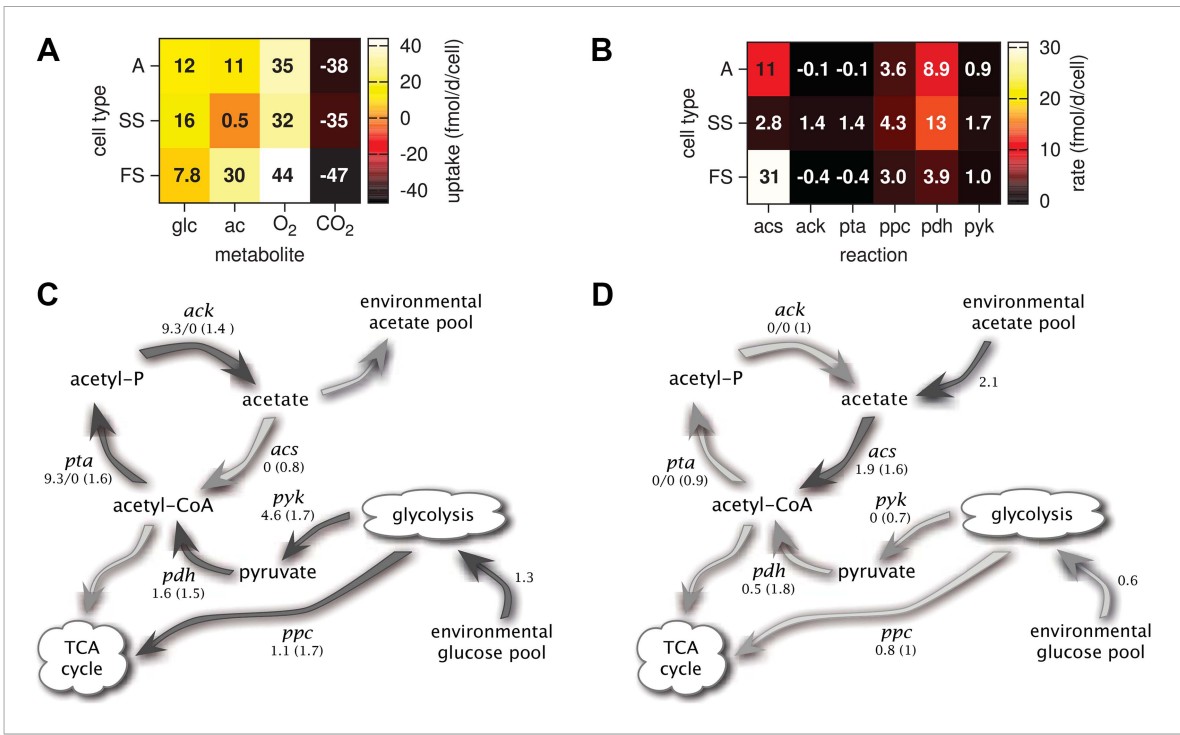

**Figure 6**. Metabolic differentiation of the A, SS and FS types. (**A**) Predicted cell-specific net metabolite uptake rates in coculture. (**B**) Predicted cell-specific reaction rates in coculture, for *acs* (acetyl-CoA synthesis), *ack* (acetate synthesis), *pta* (acetyl phosphate synthesis), *ppc* (oxaloacetate synthesis from phosphoenolpyruvate), *pdh* (decarboxylation of pyruvate to acetyl-CoA) and *pyk* (pyruvate synthesis from phosphoenolpyruvate). Rates in (**A**) and (**B**) are averaged over all time points within the first 100 days of evolution, with reversed reactions or net metabolite export represented by negative rates. (**C**) and (**D**): Simplified model subset of *E. coli* acetate and glucose metabolism, showing pathway activations in type SS (**C**) and FS (**D**) relative to type A during exponential growth in monoculture. Non-bracketed numeric values are ratios of predicted fluxes in the evolved types over fluxes in type A. Bracketed values are ratios of mRNA harvested from monoculture experiments by *Le Gac et al. (2008)*, for comparison. A ratio of 0/0 indicates zero flux in both the evolved and ancestral type, a ratio of 1 corresponds to an unchanged flux or mRNA, a ratio of 0 corresponds to complete deactivation in the evolved type. Darker arrows indicate increased predicted fluxes in the evolved type. Flux predictions correspond to the time points of mRNA measurements, that is, 3.5 hr after dilution for SS and 4 hr after dilution for A and SS (*Le Gac et al., 2008*).

catabolism to carbon dioxide is preferred over incomplete glucose catabolism with acetate excretion. On the other hand, oxygen limitation leads to an energetic tradeoff between complete glucose catabolism and efficient oxygen utilization, resulting in the excretion of acetate.

Furthermore, the depletion of oxygen during cell growth makes oxygen a temporary limiting resource for all cells (*Figure 5K*). Shortly after dilution into fresh medium, the exponential growth of the SS type on glucose leads to a rapid drop of oxygen to nanomolar concentrations. Despite oxygen diffusion into the medium, oxygen remains at sub-saturation levels for several more hours because the slow-growing acetate-consuming FS cells still consume oxygen after the growth of SS cells has halted. Differences in SS and FS growth rates (*Figure 5C,E*) thus mitigate competition for oxygen through temporal niche separation. Hence, oxygen likely plays an important role in the metabolic diversification, as previously hypothesized by *Le Gac et al. (2008)*. This shows that the splitting of metabolic pathways across specialists can be caused by the composite effects of competition for electron donors and electron acceptors.

Consistent with differential substrate usage, average cell-specific reaction rates (*Figure 6B*) reveal differences in pathway activation: The transformation of acetate into acetyl-CoA by acetyl-CoA synthetase (*acs*) is decreased in type SS and increased in type FS, when compared to the ancestral type. Furthermore, the conversion of phosphoenolpyruvate to oxaloacetate (*ppc*), the conversion of phosphoenolpyruvate to pyruvate (*pyk*) and the decarboxylation of pyruvate to acetyl-CoA (*pdh*), linking the glycolysis pathway to the citric acid cycle, are all upregulated in the SS type when compared to the FS type. Similar differences in pathway activation also exist during early exponential growth in monoculture (*Figure 6C,D*), because FS grows partly on acetate and SS excretes acetate (*Figure 4F,J*). Previous microarray profiles of mRNA concentrations during exponential growth in monocultures (*Le Gac et al., 2008*) found an upregulation of acetate consumption genes in FS and acetate excretion genes in SS compared to A, qualitatively confirming our predictions (*Figure 6C,D*). Interestingly, our simulations suggest a significant down-regulation of glucose catabolism (*pyk*, *pdh* and *ppc*) in FS compared to A, which contradicts the transcript profiles (*Figure 6D*). However, mRNA was harvested from well-aerated flasks, while the monoculture experiments (*Figure 4*) and evolution experiments (*Figure 3*) were performed in test tubes where oxygen can become limiting (*Andersen and von Meyenburg, 1980*). Oxygen becomes particularly scarce in the FS tubes (*Figure 4K*) and temporarily limits glucose catabolism, which would explain the strong downregulation not reflected in the transcript profiles (*Le Gac et al., 2008*). Furthermore, while broad pathway activation patterns could be qualitatively reproduced in our system, this might be harder in other cases due to post-transcriptional regulation or post-translational modifications (*Blazier and Papin, 2012*).

The periodic (seasonal) changes in glucose and acetate concentrations in batch culture have previously been shown to promote coexistence of the SS and FS types, in analogy to the maintenance of phytoplankton diversity via fluctuations of resource availability (*Sommer, 1984*; *Spencer et al., 2007*). Experiments with SS-FS batch cocultures revealed that the SS type quickly dominates over the FS type, when restricted to the first glucose-rich season through frequent dilution into fresh growth medium. Reciprocally, when SS and FS are grown in solution resembling the second glucose-depleted acetate-rich season, the FS type quickly dominates over the SS type (*Spencer et al., 2007*). Accordingly, in a full batch cycle the relative SS cell density has been shown to culminate within 4–8 hr and to gradually decrease afterwards (*Friesen et al., 2004*, *Figure 6B*), in consistence with our simulations (*Figure 5D*). Simulations of the SS and FS batch coculture restricted to the first or second season, analogous to Spencer et al.'s experiments, reproduce these observations and verify the role of periodic variation of glucose and acetate concentrations in maintaining the coexistence of both types (*Figure 7*, see the 'Materials and methods' for details).

## Conclusions

The models presented here make detailed predictions about the microbial dynamics in the considered experiments. First, after calibration the cell models largely explain the data from the monoculture experiments (*Figure 4*). Second, the predictions for pathway activation in the three strains (*Figure 6*) are roughly consistent with transcription profiles. Third, simulations of the microbial community consisting of all three strains (*Figure 5*) reproduce the successional dynamics of diversification observed in the evolution experiments (*Figure 3*). Fourth, simulations of the SS-FS cocultures restricted to either the glucose-rich or glucose-depleted season reproduce the dominance of the SS or FS type (*Figure 7*), respectively, in consistence with previous co-culture experiments. It is important to note that only data from monoculture experiments were used to calibrate the cell models for the

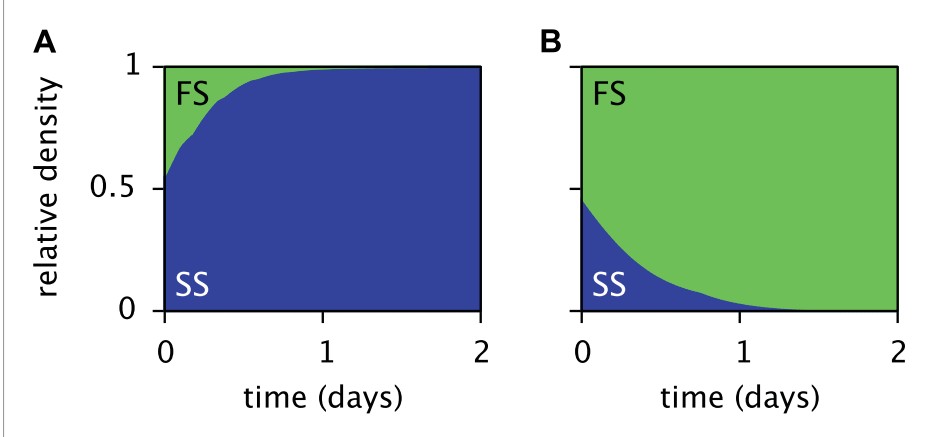

**Figure 7**. Predicted relative cell densities of the SS and FS types in batch coculture when restricted to either the first glucose-rich (**A**) or second glucose-depleted (**B**) season. In (**A**), restriction to the first season was achieved by shorter dilution periods which prevented the complete depletion of glucose. In (**B**), restriction to the second season was achieved by using the glucose-depleted acetate-rich solution, produced by the full-batch coculture, as growth medium (see the Methods for details). See *Figure 7—figure supplement 1* for results from analogous experiments by *Spencer et al. (2007)*.

The following figure supplement is available for figure 7:

**Figure supplement 1**. Measured relative cell densities of the SS and FS types in batch coculture, when restricted to either the first glucose-rich (left column) or second glucose-depleted (right column) season for three independently evolved communities (rows 1–3), as reported by Spencer *et al.* (2007, Figures 2A,B therein).

three strains (A, SS and FS). In particular, no information from co-culture experiments was used in the setup of the microbial community model, and thus there was no a priori knowledge about what the emergent community dynamics would be. Hence, our work conceptually produced non-trivial predictions that could be compared to experimental observations, although all experiments had already been performed.

Our work sheds light on the fundamental problem of metabolic diversification and the emergence of shared catabolic pathways. In particular, our model allowed quantitative predictions for the metabolic fluxes for each strain in coculture, revealing temporary cross-feeding as an emergent property of the evolved community (*Treves et al., 1998*). Cross-feeding, conventionally seen as a beneficial interaction (*Morris et al., 2013*), thus emerged as a form of niche segregation driven by competition for organic carbon and oxygen. Because both evolved types prefer glucose whenever available at high concentrations, but exchange acetate under oxygen limitation, the community constantly switches between competitive and beneficial interactions. Natural microbial populations might thus also oscillate between negative and positive interactions, for example depending on oxygen levels. Our findings also support previous suggestions that microbial evolution can be driven by deterministic ecological processes (*Wood et al., 2005*; *Oxman et al., 2008*; *Herron and Doebeli, 2013*). In this case, the observed diversification is due to competition for limiting resources whose use is constrained by basic metabolic tradeoffs. Other instances of ecological diversification in microbial evolution experiments, for example as reported by *Plucain et al. (2014)*, might be explained using a similar approach.

More generally, we have demonstrated how MCM can be used to experimentally calibrate and combine genome-based cell models to predict the emergent dynamics of microbial communities. Our framework thus provides a starting point for designing microbial communities with particular metabolic properties, such as optimized catabolic performance. While MCM is designed for genome-based metabolic models, it can also accommodate conventional functional group models. In these models, different ecological functions such as photosynthesis, heterotrophy or nitrification are performed by distinct populations whose metabolic activity is determined, for example, by Michaelis–Menten kinetics and whose growth is described by simple substrate-biomass yield factors

(*Hood et al., 2006*; *Reed et al., 2014*). Hence, natural microbial communities could be modeled even if annotated genomes are not available for each member species. While functional group models general require fewer parameters, their calibration remains a challenge (*Panikov and Sizova, 1996*). In MCM, model calibration becomes analogous to coefficient estimation in conventional multivariate regression, and can be used to estimate poorly known parameters such as stoichiometric coefficients, growth kinetics or extracellular transport coefficients (MCM user manual, *Supplementary file 1*, section 12). To our knowledge, no existing comparable framework offers the flexibility combined with the statistical functionality of MCM. In view of the increasing availability of genome-scale metabolic models (*Feist et al., 2008*), our work provides a missing link to a predictive and synthetic microbial ecology.

## Materials and methods

### MCM overview

MCM is a mathematical and computational framework for the construction, simulation, statistical analysis and calibration of microbial community models (*Figure 2*). Models are specified in special files that define all metabolites, reactions, cell species and environmental variables. MCM is controlled through custom scripts, that is, text files containing a sequence of special commands, such as for running simulations or fitting parameters. MCM includes tools for the conversion of conventional genome-scale FBA models, such as generated by the Model SEED pipeline (*Henry et al., 2010*) based on sequenced genomes, into a draft MC model.

MCM can accommodate microbial communities comprising genome-based cell models with arbitrary environmental variables, metabolite exchange kinetics and regulatory mechanisms. For example, environmental variables may be stochastic processes (e.g., representing climate), or specified using measured data (e.g., redox potential in bioreactor experiments), or depend on metabolite concentrations (e.g., pH determined by acetate concentration) or even be dynamical (e.g., temperature increasing at a rate proportional to biomass production rates). This versatility allows for the incorporation of complex environmental feedbacks, such as host immune responses in gut microbiota (*Karlsson et al., 2011*). Metabolite uptake and export rate limits can be arbitrary functions of metabolite concentrations or environmental variables. Similar interdependencies are possible for reaction rate limits, thus allowing the inclusion of inhibitory or regulatory mechanisms (*Covert et al., 2008*). Metabolite concentrations can be explicitly specified, for example, using measured time series, or depend dynamically on microbial export and other external fluxes. Effects of phage predation (*Jensen et al., 2006*), reaction energetics (*Reed et al., 2014*) or stochastic environments can also be incorporated.

MCM keeps track of a multitude of output variables such as cell densities, reaction rates, metabolite concentrations and metabolite exchange rates. Because each reaction can be formally associated with a particular enzyme, in turn encoded by a particular gene, MCM also makes predictions about gene densities as a product of cell densities and gene copy numbers per cell. Metabolic activity statistics (e.g., *Figure 6A,B*) facilitate the identification of metabolic interactions such as cross-feeding (*Morris et al., 2013*). The predicted time courses of output variables can be statistically evaluated against time series ranging from chemical concentrations, rate measurements to cell densities and metagenomics.

MC models can include arbitrary abstract (symbolic) numeric parameters with a predefined value range or probability distribution. Symbolic parameters can represent, for example, stoichiometric coefficients, gene copy numbers, cell life expectancies, half-saturation constants or environmental variables. The inclusion of symbolic parameters enables a high-level analysis of microbial communities: For example, MCM can automatically calibrate (fit) unknown symbolic parameters to time series using maximum–likelihood parameter estimation (*Eliason, 1993*). The likelihood of the data, given a particular parameter choice, is calculated by assuming a mixed deterministic-stochastic model in which the deterministic part is given by the model predictions, and the stochastic part is given by normally distributed errors. The likelihood is minimized using an iterative optimization algorithm involving step-wise parameter adjustments and repeated simulations. Other fitting algorithms are also available, such as maximization of the average coefficient of determination ($R^2$), which is equivalent to weighted least-squares fitting. Because MCM can calibrate unknown measurement units, raw uncalibrated data (e.g., optical cell densities with no calibration to colony forming units, *Figure 4A*) can also be used.

In this paper single-cell models were calibrated to monoculture experiments, however models can also be calibrated using data from experimental or natural communities that include unculturable species (MCM user manual, *Supplementary file 1*, sections 7 and 12; *Louca and Doebeli, 2015*). In general, fitted parameters need not be directly connected to the data used for calibration, as long as a change in the parameters influences the predictions that are being compared to the data. While this is a general principle of parameter estimation (*Tarantola, 2005*), in practice the uncertainty of calibrated parameters (e.g., in terms of confidence intervals) increases when their influence on the 'goodness of fit' is weaker. Moreover, alternative parameter combinations can sometimes yield a comparable match to the data, especially if multiple parameters influence the same variables (inverse problem degeneracy). Local fitting optima can be detected through repeated randomly seeded calibrations (see next section), and overfitting can be partially avoided by keeping the number of free parameters at a bare minimum. Nevertheless, in certain cases good knowledge of the system or previous literature may be required to identify the most plausible calibrations. Finally, we emphasize that MCM is, after all, merely a framework enabling the construction, calibration and analysis of microbial community models. MCM models are thus limited by the same caveats and assumptions as other constraint-based metabolic models (*Blazier and Papin, 2012*; *Antoniewicz, 2013*) and any predictions made by MCM should be subject to similar scrutiny.

## Calibration of *E. coli* cell models

*E. coli* strains were obtained from an evolution experiment performed in a batch culture environment with daily dilutions into glucose-acetate supplemented Davis minimal medium (*Spencer et al., 2008*; *Tyerman et al., 2008*). For each phenotype, three clones were isolated from population 20 after 150 days and used for three independent monoculture growth experiments. Optical densities, as well as glucose, acetate and oxygen concentration data from these experiments were used to calibrate the individual cell-metabolic models for the A, SS and FS phenotypes. Oxygen measurements were not available for type A. Experimental details and results are described by *Le Gac et al. (2008)*.

In the models, the limiting nutrients are assumed to be oxygen, glucose and acetate; all other nutrients can be taken up at an arbitrary rate. Oxygen, glucose and acetate uptake rate limits were described by Monod-like kinetics. The maximum cell-specific oxygen uptake rate was set to $1.008 \times 10^{-13} \, \mathrm{mol}/(\mathrm{d} \cdot \mathrm{cell})$, according to *Varma and Palsson (1994)*. The oxygen half-saturation constant was set to $1.21 \times 10^{-7} \, \mathrm{M}$ according to *Stolper et al. (2010)*. Oxygen was assumed to be initially at atmospheric saturation levels (0.217 mM at 37°C) and repleted at a rate proportional to its deviation from saturation (*Gupta and Rao, 2003*).

The fitted parameters for each cell type were the maximum cell-specific uptake rates and half-saturation constants for glucose and acetate, as well as initial cell densities and non-growth associated ATP maintenance energy requirements. The initial glucose and acetate concentrations were set to the average value measured at the earliest sampling point (1 hr after incubation) for each type. The oxygen mass transfer coefficient ($\mathrm{M}/\mathrm{day}$ per M deviation) was initially fitted individually for each type together with all other parameters, and then fixed to the average of all three initial fits. All other parameters were then again fitted individually for each type. Parameter fitting was done by maximizing the average coefficient of determination ($R^2$) using the MCM command *fitMCM*. A total of 237 data points were used to fit 19 parameters (Table in *Supplementary file 2*). To reduce the possibility of only reaching a local maximum, fitting was repeated 100 times for each strain starting at random initial parameter values and the best fit among all 100 runs was used. While some fitting runs reached alternative local maxima, the best overall fit was reached in most cases.

Cell densities were directly compared to optical density (OD) measurements. The appropriate calibrations were estimated by MCM and ranged within $8.2 \times 10^{11} - 1.3 \times 10^{12} \, \mathrm{cells}/(\mathrm{L} \cdot \mathrm{OD})$. These estimates are consistent with previous experimental calibrations (*Lawrence and Maier, 1977*) yielding $0.26 \, \mathrm{g} \, \mathrm{dry} \, \mathrm{weight}/(\mathrm{L} \cdot \mathrm{OD})$, which corresponds to $1.4 \times 10^{12} \, \mathrm{cells}/(\mathrm{L} \cdot \mathrm{OD})$ (assuming a cell dry weight of $1.8 \times 10^{-13} \, \mathrm{g}$ in the stationary phase; *Fagerbakke et al., 1996*).

## Simulation of the microbial community model

The microbial community model was simulated using the MCM command *runMCM*. Initial glucose and acetate concentrations were set to the average of all values measured at the earliest sampling

point of the monoculture incubations. Cell death was not explicitly included, because of lack of appropriate data for calibration and because daily dilutions by far exceeded cell death as a factor of cell population reduction.

## Robustness of the SS-FS coexistence

To verify the robustness of the stable SS-FS coexistence in coculture, we randomly varied each fitted model parameter uniformly within an interval spanning 10% above and 10% below its calibrated value. Both types coexisted in 50 out of 50 random simulations (*Figure 5—figure supplement 2*).

## Seasonal restriction of the SS-FS cocultures

Simulations of the SS-FS cocultures restricted to the first glucose-rich or second glucose-depleted season, as opposed to the full batch cycle, were performed in analogy to the experiments by *Spencer et al. (2007)*. More precisely, to model the first season experiment we changed the dilution rate to $1/32$ every 5 hr, so that at the end of each batch cycle glucose was not yet completely depleted. Similarly, for the second season experiment we changed the dilution rate to $1/32$ every 19 hr, and adjusted the growth medium to resemble the glucose-depleted acetate-rich solution reported by Spencer et al. (no glucose, 3.59 mM acetate). Initial cell densities were set to $10^{10}$ cells/l for both types. All other model parameters were kept unchanged. The original experiments by *Spencer et al. (2007)* were performed at higher dilution rates (4 and 15 hr for the first and second season experiment, respectively), however in our simulations neither the FS nor SS type could persist at these high dilution rates. We note that the strains used in our work (*Le Gac et al., 2008*) had evolved in separate evolution experiments using a different growth medium than those by *Spencer et al. (2007)*.

## Obtaining MCM

MCM is open source and available at http://www.zoology.ubc.ca/MCM.

## Acknowledgements

SL acknowledges the financial support of the PIMS IGTC for Mathematical Biology as well as the Department of Mathematics, UBC. SL and MD acknowledge the support of NSERC. We thank Evan Durno and Matthew Osmond for comments. We thank Sung Ho Yoon (Korea Research Institute of Bioscience & Biotechnology) for providing us with a copy of the REL606 cell-metabolic model (*Yoon et al., 2012*). We thank Mickaël Le Gac (Laboratoire d'Ecologie Pélagique, France) for providing us with the raw data of his monoculture incubation experiments (*Le Gac et al., 2008*).

## Additional information

### Competing interests

MD: Reviewing editor, *eLife.* The other author declares that no competing interests exist.

### Funding

| Funder | Grant reference | Author |
| --- | --- | --- |
| Natural Sciences and Engineering Research Council of Canada (NSERC) | Discovery Grant, 21990 | Michael Doebeli |
| Pacific Institute for Mathematical Sciences (PIMS) | International Graduate Training Centre (IGTC) in Mathematical Biology | Stilianos Louca |
| University of British Columbia (UBC) | Graduate Student Fellowship | Stilianos Louca |

The funders had no role in study design, data collection and interpretation, or the decision to submit the work for publication.

## Author contributions

SL, Conceived and developed MCM. Designed the *E. coli* community model and performed the calibration and simulations. Analyzed and interpreted the simulations. Wrote the article; MD, Interpreted the simulations. Contributed to writing the article. Supervised the project

## Additional files

### Supplementary files

• Supplementary file 1. A thorough user manual for MCM, including an in-depth description of MCM's mathematical framework and step-by-step examples. The latest version can also be retrieved at http://www.zoology.ubc.ca/MCM.

• Supplementary file 2. An overview of the fitted parameter values for the *E. coli* models.

• Source code 1. Includes the full calibrated MC model of the *E. coli* microbial community, together with all MCM scripts required for its simulation.

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
