## [Decision Letter]

Thank you for submitting your work entitled “Predicting the dynamics of microbial communities using genome-based metabolic models” for peer review at *eLife*. Your submission has been favorably evaluated by Naama Barkai (Senior Editor) and three reviewers, one of whom is a member of our Board of Reviewing Editors.

The reviewers have discussed the reviews with one another and the Reviewing editor has drafted this decision to help you prepare a revised submission.

This manuscript presents (a) a computational pipeline for calibrating genome-scale models of metabolism through fitting to experimental data and (b) the application of calibrated *E. coli* models to the study of a previously published evolutionary experiment, in which two mutants were found to coexist, after outcompeting the ancestral strain. What have we learned? From the point of view of this biological system, this paper provided yet another piece of data to the many already existing that the niche for FS emerges after the SS rise in frequency, but does not go to fixation. This has been known for quite a while, but it is nice to see the model agree with this finding. The calibration of individual genome scale models prior to studying communities seems a nice addition to existing frameworks, in fact one that should become a standard approach, whenever possible, for ecosystem-level modeling. The modeling and calibration software developed by the authors seems very well documented.

Shared major concerns:

1) Your paper did not explicitly compare model predictions with experiments. We would like to see a comparison of predicted metabolic and population dynamics with experimental coculture dynamics. There are no new experiments to test any predicted effect of perturbations. To at least test that FS can't invade alone is totally easy. Presumably one could also find an acetate concentration that would have been high enough to obviate the need for SS to come first (or a shorter duration or larger dilution that would have left enough acetate by the time of the transfer to have had the same effect). Another major prediction – that was pretty cool – is that acetate is excreted during a short window of time. This would be an excellent experiment to validate the model's predictions. Finally, you have to actually *show* us the comparison of predicted fluxes and mRNA that is supposed to be such a great fit. If it really is that good, we'd be rather impressed, for there is a huge literature about how this is not the case.

2) MCM seems to be sold too hard and does not responsibly acknowledge other DFBA platforms for communities like COMETS that do nearly everything mentioned here except the statistical fitting of parameters. We rather like that extension, but it seems to be advertised as more than it is.

3) Given that the authors propose their approach as broadly applicable to studying microbial communities, we think it may be important for them to comment on the realistic applicability of the method beyond *E. coli* wild type and mutants, or other well-annotated models. Would their method be useful, for example, for improving models that lack precision not just at the level of uptake kinetics, but at the level of the stoichiometry itself?

Specific major concerns:

Reviewer 1:

1) I don't understand Figure 3. If SS released acetate under oxygen limitation (∼1/4 day after daily dilution, Figure 3), then why should acetate be accumulated that quickly – immediately after daily dilution (Figure 3)?

2) The Introduction should clearly discuss where the field stands currently and how MCM pushes the field forward. I am not sure how, for example, COMETS of the Segre group could have predicted the spatial-temporal dynamics in microbial communities if they had not somehow calibrated their model. Only after I talked to an expert did I realize that usually parameter choice is done through looking for values in literature, which leaves a few tunable parameters. Then, parameters are adjusted manually to fit experimental dynamics. This is because some of these parameters may not be directly measured experimentally. A thoughtful discussion on this will be helpful.

3) The end of conclusion mentioned that parameter estimation does not necessarily require monoculture measurements. This is a critical point, and should be formally demonstrated (rather than hidden in a supplementary file). For example, the authors could model the three-member community with parameters derived from cocultures of two members starting at arbitrary initial compositions. This is to mimic cases (e.g. soil microbes) where many species are not individually culturable.

4) The flow of the paper is suboptimal, especially to an outside reader. For example, you can move “MCM overview” to immediately after the Introduction. You may also want to add concrete examples to your figures to demonstrate how MCM works in reality.

Reviewer 2:

1) The novelty of this work is mainly in the combination of different approaches and data, rather than in the approaches and data themselves. I find fascinating that the model recapitulates the observations, and that model-predicted fluxes are consistent with previously measured gene expression data. However, it is not clear to me what aspects of the insight provided by the model were not known or suspected before, given that extensive work was done on this system.

2) As mentioned above, I like the calibration approach. However, I think that some important information is missing. First, there should be a table (fine as supplementary) detailing the values of fitted parameters, and any available comparative values from the literature (only a few examples are provided in the Methods section). Also: is the fitted ATP maintenance the value for the non-growth associated maintenance, or the coefficients of growth-associated maintenance? Second, it would important for readers to know whether the solutions found by the fitting algorithm are unique, and how sensitive the result are to parameter precision. Sensitivity analyses are discussed in the user manual, and seem to have been applied to a different microbial community. I think it would be particularly important to know whether the main result of stable coexistence is sensitive to the choice of fitted parameters.

Reviewer 3:

1) The text reads that “these observations are in exact agreement with microarray transcriptional profiles”. Given the strength of “exact agreement”, I was very surprised that there was no display of experiment versus model. This was a major finding, but no figures or statistical analysis of what “exact” means.

A related point: it is claimed that “MCM also makes predictions about gene densities” because each flux is associated with an enzyme. This is actually a fairly ludicrous claim. There have been paper after paper showing that, not only can there be a lack of correlation between flux and enzyme activity, they can even be negatively correlated. This is a central tenet of Metabolic Control Analysis, and has been well documented and commented upon by folks such as Dan Fraenkel (2003, Current Opinion in Microbiology) and Hans Westerhoff (Rossell et al., 2006. PNAS). This possibility must be addressed to make the reader aware that there should never be the assumption that flux is proportionally related to enzyme levels for all enzymes. If that were so, every enzyme would have a control coefficient of 1, which is absolutely impossible because the sum of control coefficients from the entire cell is 1.

Returning to the point above, then, if there really is a good quantitative correlation between the array data from the Le Gac work and the DFBA model here – for matching timepoints as the mRNA was harvested – it would be quite a nice finding. This analysis is absolutely essential to the paper.

2) An advance stated is MCM itself. I am not sold upon exactly how this is really a major advance beyond a variety of other DFBA approaches, including prior work on communities. For example, a very large number of MCM features that are described in a way that comes across as them being novel also exist in COMETS (2014, Cell Reports). At the very least, acknowledge where this is so, and use it as an opportunity to more precisely say where this goes above and beyond. For example, I like the inherent fitting of data and perhaps more could be made of exactly how this works. COMETS does not use fitting to each substrate, and frankly an objective means to parameterize when canonical parameters fail to work well is a nice step.

[Editors' note: further revisions were requested prior to acceptance, as described below.]

Thank you for resubmitting your work entitled “Calibration and analysis of genome-based models for microbial ecology” for further consideration at *eLife*. Your revised article has been favorably evaluated by Naama Barkai (Senior Editor) and three reviewers, one of whom is a member of our Board of Reviewing Editors.

The manuscript has been improved but there are some remaining issues that need to be addressed before acceptance, as outlined below:

Reviewer 1:

For this to be a useful Tools and Resources article, I'd still like to see a discussion on MCM's limitations and requirements in the main text. For example, what is enough data for calibrating a model? Under what situations will MCM give you several solutions and what do you do in this case? What should be avoided or advocated when using MCM? You probably discussed these in the manual, but it will be helpful to summarize the main points in the text.

Reviewer 2:

The authors have overall addressed my concerns, partially by answering the issues raised, partially by transforming their work into a Tools and Resources article.

There are still two points that need some revision:

1) Even if biological insight is no more the main focus of the paper, I still think that the authors could explain a little bit in the Introduction why people may care about the specific example they study. For example, I like the following rebuttal of the authors to one of the questions: “While most of the results have been found experimentally over the course of several years, it is only now that a mechanistic model has managed to unify many of them in such a clear, unambiguous and synergistic manner. These modeling results provide very strong credence to a large body of experimental work that was done in our lab over the course of roughly a decade.” I think that a slightly expanded version of this text would help orient the broad readership towards understanding why their example is interesting (in addition to the fact that it works).

2) Most importantly, I think that in talking about their software, as well as previous tools, the authors should be very careful to clearly distinguish between features that ere possible (currently or in principle) in the different tools, vs. features that are actually presented in detail and tested in the manuscript. I am referring in particular to the last paragraph in the subsection “Model”, which is quite problematic in a number of ways. For example, regulation (both allosteric and transcriptional) in FBA models is notoriously a very tricky, overall unresolved problem. MCM has the potentially useful feature of allowing users to set rules to limit fluxes as a function of other parameters. This feature, described in one page in the user manual, is oversold in the main text as a capacity of MCM to include regulation. Similarly superficial is the description of the inclusion of phages in the model. Again, this is described shortly in the user manual with no justification or testing. All these features are unnecessarily used as a way of contrasting MCM with previous software, leading to unjustified conclusions, e.g. that COMETS “offers limited model versatility in terms of uptake kinetics” (COMETS does allow different parameters for different molecules and organisms, even if parameters were chosen to be equal in the specific simulations presented in the COMETS paper), that COMETS “seems limited to Petri dishes”, whereas “MCM can be used to understand the dynamics of realistic microbial communities, ranging from the soil or groundwater to artificial communities and bioreactors”. Both COMETS and MCM, after all, were based on the same underlying modeling framework, and tested on laboratory systems. I think the authors should definitely mention (with added accuracy) – perhaps in the Discussion – the additional features of their software that are not described and tested in detail in the main text. At the same time, I think they should limit their claims of major innovation to the components that are actually tested and presented in detail. The capacity to perform calibration on individual organisms, followed by proof of predictive capacity of the global dynamics on a highly interesting system is elegant and brilliant, and I think it is unnecessarily weakened by these other dubious claims.

Reviewer 3:

First off, I strongly agree with Reviewer 2's comments above. They have done much to improve the paper to show more and fix much of the language. As noted, however, saying why this matters in the paper as well as they did in the rebuttal would be great, and they still need to back down from hyping a large series of bells and whistles that may be great features but are not demonstrated here. There are plenty of nice advances in their method and they should stick to these.

As for my own concerns, the direct comparison to data and clarification that expression and flux should not be expected to perfectly correlate is a massive relief. The qualitative agreement is not bad in Figure 6, and can rest on its own laurels. The figures are also particularly improved in their clarity.

I still find the concept of an “average flux” over such a dynamic experiment to be strange, but at least it is now explained. Using the same time point as the RNA data is *far* better.

---

## [Author Response]

Shared major concerns:

1) Your paper did not explicitly compare model predictions with experiments. We would like to see a comparison of predicted metabolic and population dynamics with experimental coculture dynamics. There are no new experiments to test any predicted effect of perturbations. To at least test that FS can't invade alone is totally easy. Presumably one could also find an acetate concentration that would have been high enough to obviate the need for SS to come first (or a shorter duration or larger dilution that would have left enough acetate by the time of the transfer to have had the same effect).

To address this concern, we would like to first clarify our methodology: to calibrate our microbial community model we have only used data from monoculture experiments, and we have completely ignored any information from co-culture experiments. This calibration yields the parameters that we used to characterize each of the three strains (ancestor, SS and FS). We then put these three calibrated strains into an artificial MCM community, which predicted emergent dynamics of the microbial community. Because the model strains were calibrated in monoculture, there was no guarantee as to what these emergent dynamics would be. Our main result is that in fact, the emergent dynamics very closely correspond to those that had previously been found in a series of co-culture experiments. Thus, conceptually our model yielded predictions that were borne out by experiments, albeit experiments that had already been performed. Nevertheless, the editors' and reviewers' suggestion for a more extensive experimental validation is well taken, and we have now expanded that part of the manuscript.

In summary:

1) Our MCM simulations predict a clear succession of the three phenotypes and an eventual coexistence of the SS and FS phenotypes (Figure 5). This has been shown repeatedly in previous evolution experiments.

2). Our simulations of a single batch of FS and SS co-culture predict a culmination of the SS relative density within 4-8 hours, and a subsequent decline due to the growth of the FS type (Figure 5). This has been shown experimentally in Friesen et al*.* (2004, Figure 6 therein). We now discuss this observation in the manuscript (last paragraph of the “Predicting microbial community dynamics” section).

3) Our simulations of the FS and SS co-culture predict a small increase in acetate concentration, shortly after depletion of the glucose within each batch cycle (Figure 5). This has been shown experimentally by Spencer et al. (2007, p. 779). We have added an appropriate comment to the manuscript (second paragraph of the “Predicting microbial community dynamics” section).

4) Our simulations predict differential pathway activation between the SS and FS types in co-culture, particularly of the *acs*, *ack*, *pta*, *ppc*, *pdh* and *pyk* genes, which are important determinants of the acetate and glucose pathways. These patterns are qualitatively consistent with microarray transcript profiles by [37]. Also see our response to related questions by the reviewers.

5) Additional MCM simulations of the FS and SS co-culture, restricted to either the glucose-rich season or the glucose-depleted season, predict a rapid extinction of the FS or SS type, respectively (Figure 7). This behavior was found in the experiments by [53] and we now discuss this in the article (last paragraph of the “Predicting microbial community dynamics” section).

Again, it is important to point out that all the experimental findings mentioned in points 1.-5. above were obtained in experimental co-cultures, whereas our calibration involved only monocultures, so that the subsequent MCM simulations with the calibrated strains in co-culture yield dynamics that are not part of the model assumptions, but instead is an emergent property of the MCM system. The fact that this dynamics closely corresponds to various salient experimental findings in co-culture constitutes a validation of the MCM predictions.

Regarding the invasion of the FS type in the absence of SS: Simulations predict that FS also invades in the absence of type SS, however the invasion speed is much slower and FS reaches lower densities than in the presence of the SS type. This is consistent with an early presence of the FS lineage at low densities in the evolution experiments, as reported in [28], indicating that some of the first FS mutations already confer a slight advantage over the ancestor when FS is rare. We now discuss these observations in the article (first paragraph of the “Predicting microbial community dynamics” section), and we have added the appropriate figures (Figure 5—figure supplement 1 and Figure 3 in the main article).

Performing additional co-culture experiments is, unfortunately, easier said than done because these experiments were performed several years ago and the appropriately skilled staff has long left our lab. Our goal is to present a novel mathematical and computational framework, and to use our wealth of pre-existing experimental data to validate its practicality and accuracy. The fact that the experiments were done beforehand, we argue, does not influence information flow nor does it compromise the validity of our arguments.

Another major prediction – that was pretty cool – is that acetate is excreted during a short window of time. This would be an excellent experiment to validate the model's predictions.

Previous SS-FS coculture experiments (53) have indeed shown a temporary increase of acetate concentration towards the end of the SS exponential growth phase, validating our prediction. We now mention this in the article (second paragraph of the “Predicting microbial community dynamics” section).

*Finally, you have to actually* show *us the comparison of predicted fluxes and mRNA that is supposed to be such a great fit. If it really is that good, we'd be rather impressed, for there is a huge literature about how this is not the case.*

Our predicted pathway activation patterns are qualitatively consistent with the mRNA measurements by [37]. That is, the upregulation of acetate consumption genes in FS and acetate production genes in SS, when compared to the ancestor, is reflected in the transcript profiles. We now elaborate more on this comparison in the manuscript (second-last paragraph of the “Predicting microbial community dynamics” section). We also added a figure comparing the predictions to the experimental data (Figure 6). Please also see our responses to related questions by the other reviewers.

2) MCM seems to be sold too hard and does not responsibly acknowledge other DFBA platforms for communities like COMETS that do nearly everything mentioned here except the statistical fitting of parameters. We rather like that extension, but it seems to be advertised as more than it is.

We now modified our manuscript to include a better description of existing techniques (including COMETS; [25]). We also better highlight what we think are the novelties of our work. We encourage the reviewers to look at MCM's User Manual ([Supplementary-material SD2-data]) to see the versatility and wealth of functionalities introduced by MCM.

For example, COMETS offers limited model versatility in terms of uptake kinetics, no gene regulation and few environmental feedback mechanisms (namely, only dynamic extracellular metabolite concentrations). Furthermore, it assumes complete knowledge of all required model parameters and provides no generic statistical model analysis. Hence, while COMETS sets an important precedent, it is of limited applicability to microbial ecological questions outside of petri dishes.

MCM extends Harcombe et al.'s approach to more versatile microbial ecological models that include, for example, arbitrary dynamical environmental variables (e.g. pH and temperature), stochastic processes, phage predation, product inhibition, arbitrary uptake kinetics, reaction energetics and gene regulation. Furthermore, MCM can perform statistical model evaluation, sensitivity analysis, Monte Carlo-based uncertainty analysis as well as automatic parameter fitting to data. We now point out these extensions in the Model section.

*3) Given that the authors propose their approach as broadly applicable to studying microbial communities, we think it may be important for them to comment on the realistic applicability of the method beyond* E. coli *wild type and mutants, or other well-annotated models.*

While MCM is designed for genome-based metabolic models, it can also accommodate conventional functional models in which the growth of each functional group is determined, for example, by simple Michaelis-Menten kinetics (29) or the thermodynamics of biocatalyzed redox reactions (49). Hence, natural microbial communities could be modeled even if annotated genomes are not available for each member species. While functional models require fewer parameters, their calibration remains a challenge (47). MCM would thus also be useful to such conventional modeling efforts. We now discuss this in the Conclusions.

Would their method be useful, for example, for improving models that lack precision not just at the level of uptake kinetics, but at the level of the stoichiometry itself?

Thanks to MCM's design, any numerical model parameter can in principle be calibrated, provided that suitable data is available. That includes stoichiometric coefficients, uptake kinetics, extracellular transport coefficients (e.g. for O_2_, as in our model), cell death rates, product inhibition kinetics or environmental pH levels. We now make this more clear in the Conclusions and the Methods.

Specific major concerns:

Reviewer 1:

*1) I don't understand*
Figure 3*. If SS released acetate under oxygen limitation (∼1/4 day after daily dilution,*
Figure 3
*i,j), then why should acetate be accumulated that quickly – immediately after daily dilution (*Figure 3*)?*

Acetate is included in the growth medium with every dilution (see the Methods for details). In fact, the acetate produced by the SS type is only a small fraction of the total acetate present. We now make this more clear in section “Experimental calibration”, by emphasizing that the dilution is performed “into fresh glucose-acetate supplemented medium”. Please note that Figure 3 is now renamed to Figure 5.

*2) The Introduction should clearly discuss where the field stands currently and how MCM pushes the field forward*.

We now added the following to the Introduction:

“So far, the standard approach has been to obtain each parameter through laborious specific measurements or from the available literature, or to manually adjust parameters to match observations (39; 7; 25). Furthermore, statistical model evaluation and sensitivity analysis is typically performed using ad-hoc code, thus increasing the effort required for the construction of any new model.”

We now also moved the Model section so that it immediately follows the Introduction. There we explain the underlying concepts and clarify our contributions to the field of microbial community modeling (last paragraph).

*I am not sure how, for example, COMETS of the Segre group could have predicted the spatial-temporal dynamics in microbial communities if they had not somehow calibrated their model*.

Segre's group chose uptake kinetic parameters, as well as diffusion coefficients, to be equal for all substrates and based on typical values from the literature. Not doubting the quality of their work, we are unsure about the biological realism of such assumptions, but we prefer not to discuss this in the paper. Certainly, in many other cases a differentiation of kinetics between different metabolites and a calibration of parameters will be needed. Currently, no systematic automated method exists for estimating arbitrary model parameters.

*Only after I talked to an expert did I realize that usually parameter choice is done through looking for values in literature, which leaves a few tunable parameters. Then, parameters are adjusted manually to fit experimental dynamics. This is because some of these parameters may not be directly measured experimentally. A thoughtful discussion on this will be helpful*.

We now discuss this in the Introduction and the Conclusions. In the Model section (last paragraph) we further elaborate on the advancements of MCM compared to COMETS.

*3) The end of conclusion mentioned that parameter estimation does not necessarily require monoculture measurements. This is a critical point, and should be formally demonstrated (rather than hidden in a supplementary file). For example, the authors could model the three-member community with parameters derived from cocultures of two members starting at arbitrary initial compositions. This is to mimic cases (e.g. soil microbes) where many species are not individually culturable*.

We designed MCM so that it is completely irrelevant whether the fitted parameters belong to a one- or multiple-species model. Including an additional example in the main article demonstrating this would, in our opinion, not add anything new conceptually, neither is it required for the completeness of our arguments. In fact, as we explain above, our goal was to demonstrate that independent data from monoculture experiments (i.e. without knowledge of the dynamics emerging in co-culture) can be used to calibrate a model, which would then predict novel emergent dynamics of a microbial community.

We do realize that some readers might be skeptical about MCM's ability to calibrate multi-species models. Hence, we now specify the section in the supplement where the interested reader can see an elaborate example using real data. We are reluctant to include that example in the main article, because it would confuse our main example of the *E. coli* evolution experiment. We would be happy to create a separate supplementary section focused on such an example, if the editors or reviewers believe that is appropriate.

We have recently completed a separate project where we use MCM to study the stability of microbial communities in bioreactors. That work is based on calibration of a multi-species model to real data, and has recently been submitted for review in another journal.

*4) The flow of the paper is suboptimal, especially to an outside reader. For example, you can move “MCM overview” to immediately after the Introduction*.

We have now restructured the manuscript in line with the reviewer's suggestions. We now introduce the model framework (in particular FBA and DFBA) and MCM's principles right after the Introduction. We then introduce the microbial evolution experiments. We still keep a more technical “MCM overview” section in the Methods.

*You may also want to add concrete examples to your figures to demonstrate how MCM works in reality*.

We have now added appropriate comments to the MCM overview figure and its legend (Figure 2 in the revised manuscript). We hope that the Methods section and our extensive MCM User Manual ([Supplementary-material SD2-data]) will provide sufficient information to the interested reader.

Reviewer 2:

*1) The novelty of this work is mainly in the combination of different approaches and data, rather than in the approaches and data themselves. I find fascinating that the model recapitulates the observations, and that model-predicted fluxes are consistent with previously measured gene expression data. However, it is not clear to me what aspects of the insight provided by the model were not known or suspected before, given that extensive work was done on this system*.

Our main contribution is the computational framework, MCM, which combines microbial community model construction with statistical model evaluation, sensitivity analysis and automated, statistically rigorous calibration to data. The versatility and functionalities of MCM are discussed and demonstrated throughout our paper. We now elaborate more on the novelties of MCM compared to existing techniques (e.g. in the Introduction and the Model sections).

Furthermore, to the best of our knowledge, this is the first time that monocultures of multiple strains were used to calibrate individual FBA models, which were then combined into a microbial community to predict the dynamics emerging in co-culture.

We go through some lengths to show that this approach, and MCM for that matter, is capable of correctly predicting the dynamics of microbial communities, both in terms of population dynamics as well as pathway activation patterns and metabolic fluxes (e.g. periodic acetate production by SS). In view of the conventional approach for obtaining model parameters (see Introduction), we expect that systematic model calibration as demonstrated in our paper will find widespread use in microbial systems biology.

In terms of the results on the *E. coli* system, we agree that our paper is mainly a synthesis. While most of the results have been found experimentally over the course of several years, it is only now that a mechanistic model has managed to unify many of them in such a clear, unambiguous and synergistic manner. These modeling results provide very strong credence to a large body of experimental work that was done in our lab over the course of roughly a decade.

To conclude, as we focus mainly on methodological novelties and synthesis, our paper is better suited as a Tools and Resources paper. As suggested, we have adjusted our title accordingly.

*2) As mentioned above, I like the calibration approach. However, I think that some important information is missing. First, there should be a table (fine as supplementary) detailing the values of fitted parameters, and any available comparative values from the literature (only a few examples are provided in the Methods section). Also: is the fitted ATP maintenance the value for the non-growth associated maintenance*, *or the coefficients of growth-associated maintenance?*

The fitted ATP maintenance is the non-growth associated maintenance, in terms of ATP molecules per unit time. The appropriate “loss reaction” is part of the *E. coli* FBA model template, published by [63]. Growth-associated maintenance is incorporated as inefficiencies of biomass synthesis from precursors biomolecules. We now make that more clear in the Methods section. We have also included a table listing the fitted parameters (Supplementary file 3), as suggested by the reviewer.

The value of non-growth associated maintenance energy was calibrated, along with the other model parameters, using the available monoculture data. For clarification, in general fitted parameters need not be directly connected to the data used for calibration, as long as a change of the parameters influences the variables that are being compared to the data. This is a general principle of parameter estimation, which is further exemplified in the MCM user manual (e.g. section 2.6). However, the uncertainty of calibrated parameters increases when their influence on the “goodness of fit” is weaker.

*Second, it would important for readers to know whether the solutions found by the fitting algorithm are unique, and how sensitive the result are to parameter precision. Sensitivity analyses are discussed in the user manual, and seem to have been applied to a different microbial community. I think it would be particularly important to know whether the main result of stable coexistence is sensitive to the choice of fitted parameters*.

To reduce the possibility of only reaching a local maximum, fitting was repeated 100 times for each strain starting at random initial parameter values and the best fit among all 100 runs was used (MCM handles this automatically). While some fitting runs reached alternative local maxima, the best overall fit was reached in most cases. We have now added this information to the Methods section.

We also analyzed the sensitivity of stable SS-FS coexistence in co-culture by randomly choosing each fitted parameter within an interval spanning 10% below and 10% above its calibrated value. In 50 out of 50 random simulations both types coexisted. We now mention this in the Methods section and have added an appropriate figure to the supplement (Figure 5—figure supplement 2).

Reviewer 3:

*1) The text reads that “these observations are in exact agreement with microarray transcriptional profiles”. Given the strength of “exact agreement”, I was very surprised that there was no display of experiment versus model. This was a major finding, but no figures or statistical analysis of what “exact” means*.

FBA makes predictions about reaction rates, and a correlation with mRNA levels is expected to be at most qualitative, not quantitative.

Our choice of wording was unfortunate, and we now corrected it to emphasize the qualitative nature of the agreement. That is, the upregulation of acetate consumption genes in FS and acetate production genes in SS, when compared to the ancestor, is reflected in the mRNA profiles. We now elaborate this comparison in the manuscript (at the end of the “Predicting microbial community dynamics” section). We also added a figure comparing the predictions to the experimental data (Figure 6).

*A related point: it is claimed that “MCM also makes predictions about gene densities” because each flux is associated with an enzyme. This is actually a fairly ludicrous claim. There have been paper after paper showing that, not only can there be a lack of correlation between flux and enzyme activity, they can even be negatively correlated. This is a central tenet of Metabolic Control Analysis, and has been well documented and commented upon by folks such as Dan Fraenkel (2003, Current Opinion in Microbiology) and Hans Westerhoff (Rossell et al., 2006. PNAS). This possibility must be addressed to make the reader aware that there should never be the assumption that flux is proportionally related to enzyme levels for all enzymes. If that were so, every enzyme would have a control coefficient of 1, which is absolutely impossible because the sum of control coefficients from the entire cell is 1*.

We appreciate the reviewer's (correct) observation. To clarify, we do not assume that flux is correlated with enzyme activity in any way. Predictions about gene densities (i.e. gene copies per liter) made by MCM are based on predicted cell densities and the number of gene copies per cell. Hence, for example, the density of the *amo* gene (associated with aerobic ammonium oxidation) would be the sum of cell densities for all cell species capable of producing the *amo* enzyme (i.e., in the context of FBA, capable of potentially performing the *amo* reaction) multiplied by the number of *amo* gene copies per cell. Gene copies can be different across cell species. This is explained in detail in the MCM user manual. We now also make that more clear in the manuscript (“MCM overview” section in the Methods).

Of course, the very assumption that each reaction corresponds to one gene via one enzyme can be questioned, e.g. when protein complexes are encoded by several genes. Hence, the final interpretation or use of MCM's predictions about gene densities should be case-dependent and subject to the user's judgment.

*Returning to the point above, then, if there really is a good quantitative correlation between the array data from the Le Gac work and the DFBA model here – for matching timepoints as the mRNA was harvested – it would be quite a nice finding. This analysis is absolutely essential to the paper*.

We now elaborate on that comparison in more detail. We also added a figure, which directly compares our predictions with Le Gac's results (Figure 6). To clarify, we do not claim a quantitative correlation exists between reaction rates and mRNA concentrations. Rather, our predictions on the upregulation of acetate production genes in SS and acetate consumption genes in FS, when compared to the ancestor, are consistent with analogous changes in mRNA levels. We now make that more clear.

*2) An advance stated is MCM itself. I am not sold upon exactly how this is really a major advance beyond a variety of other DFBA approaches, including prior work on communities. For example, a very large number of MCM features that are described in a way that comes across as them being novel also exist in COMETS (2014, Cell Reports). At the very least, acknowledge where this is so, and use it as an opportunity to more precisely say where this goes above and beyond. For example, I like the inherent fitting of data and perhaps more could be made of exactly how this works. COMETS does not use fitting to each substrate, and frankly an objective means to parameterize when canonical parameters fail to work well is a nice step*.

As mentioned in our responses to Reviewer 1, in contrast to COMETS, MCM can accommodate more versatile microbial ecological models that include, for example, dynamical environmental variables (such as pH or temperature, that can change in response to microbial metabolism), stochastic processes, phage predation, product inhibition, arbitrary uptake kinetics, reaction energetics and gene regulation. Furthermore, MCM can perform statistical model evaluation and sensitivity analysis, as well as automatic parameter fitting to data.

In the Model section (last two paragraphs) we now compare our work to previous DFBA approaches, and in particular COMETS, and we further emphasize the importance of MCM's fitting capacities in the Conclusions section. We now also elaborate more on the fitting algorithms used by MCM in the Methods section.

[Editors' note: further revisions were requested prior to acceptance, as described below.]

Reviewer 1:

For this to be a useful Tools and Resources article, I'd still like to see a discussion on MCM's limitations and requirements in the main text. For example, what is enough data for calibrating a model? Under what situations will MCM give you several solutions and what do you do in this case? What should be avoided or advocated when using MCM? You probably discussed these in the manual, but it will be helpful to summarize the main points in the text.

The amount of required data is highly case-specific and indeed a non-trivial question that cannot be answered in general. MCM calibrations will always be subject to the same fundamental limitations as other non-linear inverse problems. We now added a discussion of the points raised by the reviewer, in the “MCM overview” section: “In this paper single-cell models were calibrated to monoculture experiments […] MCM models are thus limited by the same caveats and assumptions as other constraint-based metabolic models (3; 2) and any predictions made by MCM should be subject to similar scrutiny.”

Reviewer 2:

*The authors have overall addressed my concerns, partially by answering the issues raised, partially by transforming their work into a Tools and Resources*.

There are still two points that need some revision:

*1) Even if biological insight is no more the main focus of the paper, I still think that the authors could explain a little bit in the Introduction why people may care about the specific example they study. For example, I like the following rebuttal of the authors to one of the questions: “While most of the results have been found experimentally over the course of several years, it is only now that a mechanistic model has managed to unify many of them in such a clear, unambiguous and synergistic manner. These modeling results provide very strong credence to a large body of experimental work that was done in our lab over the course of roughly a decade.” I think that a slightly expanded version of this text would help orient the broad readership towards understanding why their example is interesting (in addition to the fact that it works)*.

We now finish the Introduction with the following passage: “To demonstrate the potential of MCM, we modeled a bacterial community that has emerged from in-vitro evolution experiments […] a large body of experimental work that was done in our lab over the course of roughly a decade.”

*2) Most importantly, I think that in talking about their software, as well as previous tools, the authors should be very careful to clearly distinguish between features that ere possible (currently or in principle) in the different tools, vs. features that are actually presented in detail and tested in the manuscript. I am referring in particular to the last paragraph in the subsection “Model”, which is quite problematic in a number of ways. For example, regulation (both allosteric and transcriptional) in FBA models is notoriously a very tricky, overall unresolved problem. MCM has the potentially useful feature of allowing users to set rules to limit fluxes as a function of other parameters. This feature, described in one page in the user manual, is oversold in the main text as a capacity of MCM to include regulation. Similarly superficial is the description of the inclusion of phages in the model. Again, this is described shortly in the user manual with no justification or testing. All these features are unnecessarily used as a way of contrasting MCM with previous software, leading to unjustified conclusions, e.g. that COMETS “offers limited model versatility in terms of uptake kinetics” (COMETS does allow different parameters for different molecules and organisms, even if parameters were chosen to be equal in the specific simulations presented in the COMETS paper), that COMETS “seems limited to Petri dishes”, whereas “MCM can be used to understand the dynamics of realistic microbial communities, ranging from the soil or groundwater to artificial communities and bioreactors”. Both COMETS and MCM, after all, were based on the same underlying modeling framework, and tested on laboratory systems. I think the authors should definitely mention (with added accuracy) – perhaps in the Discussion – the additional features of their software that are not described and tested in detail in the main text. At the same time, I think they should limit their claims of major innovation to the components that are actually tested and presented in detail. The capacity to perform calibration on individual organisms, followed by proof of predictive capacity of the global dynamics on a highly interesting system is elegant and brilliant, and I think it is unnecessarily weakened by these other dubious claims*.

Regarding gene regulation:

Incorporating post-transcriptional or post-translational gene regulation in FBA models is indeed tricky for fundamental reasons, and we likely have overstated MCM's potential to that end. The incorporation of gene regulation into MCM models follows published regulatory FBA approaches ([10]; [9]; [8]; Herrgard et al., 2006) that have been shown to improve the fidelity of *E. coli* and *S. cerevisiae* FBA models, but are yet to be tested at the microbial community level. Hence, the underlying theory is definitely not our innovation, and MCM merely provides a convenient tool for constructing and calibrating such models. We now clarify this in the last paragraph of the “Model” section. We also further explain regulatory FBA and non-stationary cell models in the MCM User Manual.

Regarding COMETS:

We would like to clarify that while COMETS does indeed allow different parameters for different molecules and organisms, the assumed kinetics are all the same, namely single-substrate Michaelis-Menten uptake kinetics (with an optional Hill coefficient) and constant reaction rate bounds. More complicated kinetics, or multi-substrate dependencies, or environmental influences (such as effects of temperature or pH) are simply not possible. We have now rephrased our conclusions on COMETS to the following: “Hence, while COMETS sets an important precedent, considerable work is still needed to make DFBA a practical approach in microbial ecosystem modeling.” We have also adjusted the remaining text in that paragraph (last paragraph of the “Model” section), where we explain the main distinguishing features of MCM.

Regarding MCM's untested features:

At the end of the day MCM is a framework facilitating the construction, analysis and calibration of microbial ecosystem models. MCM's versatility should thus be understood in terms of possible model structures and possible couplings between components (e.g. environment-cell coupling). The actual model will always be subject to the user's choice. Hence, for example, saying that MCM “supports phage predation” is merely a claim that MCM supports model structures that have conventionally been used to model phage predation (at the population level; for an example see [31]). Similarly, when we say “MCM supports reaction energetics” this means that MCM allows for the calculation of free energies and their incorporation into models (e.g. influencing growth factors), regardless of how this is used in any particular case (for an example see [49]).

We now make this clear in the “MCM overview” section, saying: “Finally, we emphasize that MCM is, after all, merely a framework enabling the construction, calibration and analysis of microbial community models. MCM models are thus limited by the same caveats and assumptions as other constraint-based metabolic models (3; 2) and any predictions made by MCM should be subject to similar scrutiny.”

Furthermore, to avoid confusion, we omit several of MCM's features from the main text. For the ones kept, we refer to existing literature that demonstrates their potential use.

Reviewer 3:

*First off, I strongly agree with Reviewer 2's comments above. They have done much to improve the paper to show more and fix much of the language. As noted, however, saying why this matters in the paper as well as they did in the rebuttal would be great, and they still need to back down from hyping a large series of bells and whistles that may be great features but are not demonstrated here. There are plenty of nice advances in their method and they should stick to these*.

We have now further emphasized the significance of our experimental system in the Introduction. We also rephrased the comparison of MCM to previous tools in a more conservative way. Please see our responses to Reviewer 2 for details.

*As for my own concerns, the direct comparison to data and clarification that expression and flux should not be expected to perfectly correlate is a massive relief. The qualitative agreement is not bad in*
Figure 6*, and can rest on its own laurels. The figures are also particularly improved in their clarity*.

*I still find the concept of an “average flux” over such a dynamic experiment to be strange, but at least it is now explained. Using the same time point as the RNA data is* far *better*.

For the comparison of predicted pathway activations to transcript microarray profiles (Figure 6) we do indeed use the same time points as the RNA data. We now make that more clear by ending the figure caption as follows: “Flux predictions correspond to the time points of mRNA measurements, i.e. 3.5 hours after dilution for SS and 4 hours after dilution for A and SS (37).”